# Virulence Factor Genes in Invasive *Escherichia coli* Are Associated with Clinical Outcomes and Disease Severity in Patients with Sepsis: A Prospective Observational Cohort Study

**DOI:** 10.3390/microorganisms11071827

**Published:** 2023-07-17

**Authors:** Valentino D’Onofrio, Reinoud Cartuyvels, Peter E. A. Messiaen, Ivan Barišić, Inge C. Gyssens

**Affiliations:** 1Faculty of Medicine and Life Sciences, Hasselt University, Martelarenlaan 42, 3500 Hasselt, Belgium; peter.messiaen@jessazh.be; 2Department of Infectious Diseases and Immunity, Jessa Hospital, 3500 Hasselt, Belgium; 3Department of Internal Medicine and Radboud, Center for Infectious Diseases, Radboud University Medical Center, Geert Grooteplein Zuid 10, 6525 GA Nijmegen, The Netherlands; 4Department of Clinical Biology, Jessa Hospital, 3500 Hasselt, Belgium; reinoud.cartuyvels@jessazh.be; 5Austrian Institute of Technology, 1210 Vienna, Austria; ivan.barisic@ait.ac.at

**Keywords:** virulence factors, *E. coli*, sepsis, clinical outcome, whole genome sequencing

## Abstract

Background: *Escherichia coli* harbours virulence factors that facilitate the development of bloodstream infections. Studies determining virulence factors in clinical isolates often have limited access to clinical data and lack associations with patient outcome. The goal of this study was to correlate sepsis outcome and virulence factors of clinical *E. coli* isolates in a large cohort. Methods: Patients presenting at the emergency department whose blood cultures were positive for *E. coli* were prospectively included. Clinical and laboratory parameters were collected at admission. SOFA-score was calculated to determine disease severity. Patient outcomes were in-hospital mortality and ICU admission. Whole genome sequencing was performed for *E. coli* isolates and virulence genes were detected using the VirulenceFinder database. Results: In total, 103 *E. coli* blood isolates were sequenced. Isolates had six to 41 virulence genes present. One virulence gene, *kpsMII_K23*, a K1 capsule group 2 of *E. coli* type K23, was significantly more present in isolates of patients who died. *kpsMII_K23* and *cvaC* (Microcin C) were significantly more frequent in isolates of patients who were admitted to the ICU. Fourteen virulence genes (*mchB*, *mchC*, *papA_fsiA_F16*, *sat*, *senB*, *iucC*, *iutA*, *iha*, *sfaD*, *cnf1*, *focG*, *vat*, *cldB*, and *mcmA)* significantly differed between patients with and without sepsis. Conclusions: Microcins, toxins, and fimbriae were associated with disease severity. Adhesins and iron uptake proteins seemed to be protective. Two genes were associated with worse clinical outcome. These findings contribute to a better understanding of host-pathogen interactions and could help identifying patients most at risk for a worse outcome.

## 1. Introduction

*Escherichia coli* lives as a commensal bacterium in animal and human intestines and can also behave as a pathogen that can cause life-threatening diseases. Due to its clinical importance, eight phylogroups are defined which can be roughly linked to lifestyle of *E. coli* [1]. Commensal strains are mostly part of phylogroup A and rarely cause disease. When bacteria acquire virulence attributes, however, they become more pathogenic. Therefore, *E. coli* can be classified into three categories: commensal, intestinal pathogenic and extra-intestinal pathogenic *E. coli* (ExPEC) [1].

*E. coli* is the major causative pathogen of intra-abdominal infections, urinary tract infections (UTI), and bloodstream infections (BSI) [2]. The latter two are caused by ExPEC and are named UroPathogenic *E. coli* (UPEC) and Sepsis Pathogenic *E. coli* (SePEC) [1] and are most frequently part of phylogroup B2 and D [1,2,3]. Phylogroup B2 contains *E. coli* that frequently accumulate virulence factors (VF), and strains have the greatest diversity of all groups [1,3].

Recent literature suggests that strains acquire new VF mostly from horizontal gene transfer. Although VF can be part of plasmids, many are incorporated in the bacterial chromosome [1]. Typically, VF genes cluster together in the chromosome on so-called pathogenicity islands (PAI) and can therefore also be easily transferred [1]. PAIs are large, horizontally transferable genomic elements that play an important role in the evolution of pathogenic *E. coli* [3]. They are non-replicative and lack the ability to self-mobilize. PAIs are not present in non-pathogenic strains. PAIs have a biased gene sequence with different G + C content in comparison to the rest of the genome and have gene or motif contents such as tRNA, direct repeats, integrases, and mobility-related genes [1]. Biggel et al. recently demonstrated in a genome-wide association study that PAIs containing the *papGII* gene (P-fimbriae encoding) have led to an emergence of invasive UPEC lineages that are more frequently seen in patients with urosepsis than in patients with cystitis [3].

ExPEC encode VF that help colonize the digestive tract and then move on to the urinary tract. They use adhesins to adhere to host cells, invasion factors and toxins to stimulate bacterial internalization. In some cases, they use iron uptake systems for iron acquisition in urine and blood and protectins to protect them from the host’s immune mechanisms [1]. A recent review by Desvaux et al. provides an excellent overview of the most important and most studied VF for UPEC and SePEC strains (see Appendix A).

Most of these studies analysing VF used PCR-based techniques to characterize extended spectrum beta-lactamase (ESBL)-producing, multidrug-resistant (MDR) organisms. Since the inclusion of MDR pathogens results in an already biased preselection of VF and host outcome, a study on isolates from a less pre-conditioned population would be beneficial to assess the relative contribution of VF to disease severity.

Overall, studies assessing VF in isolates from clinical samples often have limited access to clinical findings. Most variables are limited to demographics and comorbidities. Although these are important confounding factors, they do not completely describe the host response. A correlation of virulence profile with host response mechanisms (clinical and laboratory parameters and inflammatory biomarkers) could provide a clearer picture of the association of VF with disease severity. Therefore, the goal of this study was to evaluate the VF profile of *E. coli* isolated from patients admitted at the emergency department and correlate this with the clinical course and outcome of their infection.

## 2. Methods

### 2.1. Literature Search

An exploratory literature search was performed in February 2021. The search terms “Invasive *E. coli* and virulence” yielded 1780 hits, starting from 1966. Therefore, the search terms were further specified. “Invasive *E. coli* and Virulence and bacteraemia” provided 86 results. Some of the most cited and most recent papers are listed in Appendix A.

### 2.2. Design and Patients of the FAPIC Study

This project is part of a prospective observational cohort study which included patients between February 2019 and April 2020 at the Jessa hospital, Hasselt, a 981-bed teaching hospital (clinicaltrial.gov identifier NCT03841162). Adult patients presenting at the emergency department (ED), the department of infectious diseases/nephrology, or the department of haemodialysis with suspected sepsis and for whom blood cultures were drawn, were asked to participate in the study. Patients were included after collection of the first set of blood cultures. Patients could be included multiple times if they developed a new suspected sepsis episode. A new episode was defined as a minimal interval of seven days between positive cultures with the same pathogen or at least 24 h between positive cultures with different organisms from the same site. This is a sub-study in which all patients with proven *E. coli* bacteraemia, i.e., blood cultures positive for *E. coli*, were included.

### 2.3. Ethical Considerations

All procedures performed in studies involving human participants were in accordance with the ethical standards of the institutional and/or national research committee and with the 1964 Helsinki Declaration and its later amendments or comparable ethical standards. Documented approval for the FAPIC study was obtained from the Ethics committees of Hasselt University and Jessa Hospital (18.106/infect18.03 and 19.51/infect.19.02). Written informed consent was obtained from all participants. Bacterial isolates were collected as part of the FAPIC prospective observational cohort study. This sub-analysis was approved by the Ethics committees of Hasselt University and Jessa Hospital (2021/021).

### 2.4. Microbiological Diagnostics and Bacterial Isolates

Blood cultures were performed for all patient episodes using the BACTEC FX (Becton Dickinson, Franklin Lakes, NJ, USA) system. Bacterial identification was done by MALDI-TOF Biotyper (Bruker, Billerica, MA, USA). Susceptibility testing was done by the Phoenix system TM 100 (Becton Dickinson). Blood cultures were processed 24 h/day, 7 days/week. All *E. coli* isolates were stored in standard culture media at room temperature (15–25 °C) for maximally two years, until the end of the study, before batch analysis. After storage, the viability of isolates was checked by re-culturing bacteria. Other clinical microbiological diagnostics were performed if deemed relevant by the treating physician. This included cultures of urine, the lower respiratory tract, and samples of specific foci.

### 2.5. Patient Data Collection

Clinical and laboratory parameters were collected at the start of each new episode from patients’ electronic medical files. ED physicians ordered clinical, biochemical, and microbiological tests guided by a suspected sepsis protocol in place at the ED. Clinical parameters included body temperature, heart rate, mean arterial pressure (MAP), oxygen saturation (SaO_2_) and partial oxygen pressure (PaO_2_), Glasgow Coma scale (GCS), the presence of central lines at admission, vasopressor use, and oxygen requirements. Laboratory testing included white blood cell count (WBC), platelet count, haemoglobin, red blood cell distribution width (RDW), c-reactive protein (CRP), creatinine, urea, lactate dehydrogenase (LDH), bilirubin, alanine aminotransferase (ALT), and aspartate aminotransferase (AST). Additional biochemical tests ordered as part of the FAPIC study were serum lactate and ferritin, based on recent insights regarding their association with sepsis mortality [4,5]. (SOFA) score was calculated for all patients [4,5]. Laboratory parameters that were not ordered by the ED physician were retrospectively determined on the same samples to reduce missing values. Recorded patient outcomes were in-hospital mortality, intensive care unit (ICU) admission (at any time during hospital admission), length of stay (LOS) both at the hospital and in the ICU. Patients were followed until hospital discharge or in-hospital death. Thus, no patients were lost to follow-up. The presence of bacteraemia was determined based on positive blood cultures that were classified as true bacteraemia or as contamination according to CDC guidelines [6]. Positive blood cultures with skin flora, such as coagulase-negative staphylococci, were considered as contaminated when less than two blood culture bottles from one patient were positive for skin flora. Patients with a positive blood culture with these organisms and a clinical suspicion of an infection of central venous catheters or surgically implanted prosthetic material were considered to have true bacteraemia. A definitive diagnosis (UTI, BSI, or abdominal infection) was made by chart review and pre-defined definitions by an experienced physician (I.C.G.) who was not involved in the care of patients, to minimize bias. Sepsis was defined as an increase in Sequential Organ Failure Assessment (SOFA) score of 2 points or more from baseline, based on the Sepsis-3 guidelines [5].

### 2.6. Whole Genome Sequencing

Isolates were transported to the Molecular Diagnostics group at the Austrian Institute of Technology where they were checked for viability at arrival. Fifteen isolates were not viable after transportation, and sequencing failed in three isolates. After a viability check of all isolates, genomic DNA was extracted with the QiAmp DNA mini kit (Qiagen, Hilden, Germany). Whole genome sequencing (WGS) was performed using the Ion Torrent PGM platform using 400 bp read chemistry. Sequencing was performed according to the protocol recommended by Life Technologies. The Ion Xpress Plus Fragment Library Kit was used to enzymatically shear 100 ng of the genomic DNA. The target fragment size was 400 bp. Subsequently, the fragmented DNA was processed using the Ion DNA Barcoding kit (Life Technologies, Carlsbad, CA, USA) and its size selected using the E-Gel SizeSelect 2% Agarose kit (Life Technologies). The size distribution of the DNA fragments was analysed using the High Sensitivity Kit (Agilent, Santa Clara, Santa Clara, CA, USA). Further sample processing was performed using the Ion OneTouch Kit (Life Technologies). Finally, the amplified DNA was sequenced using the 318 chip (Life Technologies). Raw reads were assembled de novo using Assembler SPAdes software [7]. The genome was annotated using the RAST (Rapid Annotations using Subsystems Technology) database [8,9].

### 2.7. Control Samples

Twenty sequences from *E. coli* isolates from clinical specimens other than blood were randomly selected from the NCBI Genbank. Virulence genes were detected using the same methods. Virulence genes were compared using X^2^ or Fisher’s Exact test between control sequences and clinical isolates.

### 2.8. Bioinformatic Analyses

Multilocus sequence typing (MLST), serotyping, and plasmid replicon typing were performed using the tools from the Center for Genomic Epidemiology website [10,11]. Antibiotic resistance genes were searched using the ResFinder database (version 23 September 2021) [12]. The VirulenceFinder database (version 29 May 2021) was used for virulence genes detection. Accession numbers of all significantly different virulence genes were used for BLAST analysis to search for previously identified blood isolates with the same virulence genes. Genetic diversity of clinical isolates was analysed using the iNext R-package.

### 2.9. Statistical Analyses

Descriptive statistics were used to analyse patient characteristics. Continuous data (median (interquartile range (IQR)) and Categorical data (number and proportion) are reported. Kappa agreement was performed to assess the agreement between genotypic and phenotypic antibiotic resistance of isolates, and the following criteria were used: k < 0 reflects ‘poor’, 0 to 0.20 ‘slight’, 0.21 to 0.4 ‘fair’, 0.41 to 0.60 ‘moderate’, 0.61 to 0.8 ‘substantial’, and above 0.81 ‘almost perfect’. X^2^ or Fisher’s Exact test (categorical) was used for univariate analyses to compare virulence genes between clinical isolates and controls and between clinical isolates with different outcomes (mortality, ICU admission, the presence of sepsis, or source of infection). A *p*-value of <0.05 was considered statistically significant. Unadjusted relative risk with a 95% confidence interval was calculated. Additionally, the absolute risk was calculated using baseline absolute risk reported by the WHO [13]. Adjustments for age, sex or disease severity were not performed.

### 2.10. Role of the Funding Source

Neither the funding source nor the Limburg Clinical Research Centre had any role in the study design; the collection, analysis, and interpretation of data; the writing of the report; and in the decision to submit the paper for publication.

## 3. Results

### 3.1. Patient Demographics

In total, 121 *E. coli* blood isolates from 113 patients were collected. Diagnoses of infection, SOFA scores, and patient outcomes are shown in Table 1. There were no missing data.

### 3.2. Clinical Isolate Characteristics

In total, 103 isolates were sequenced. Appendix A shows the distribution of MLST and serotypes identified in isolates. MLST number 73 and 69 and serotype O6:H1 were the most frequent in 18 and 15 isolates, respectively. Several plasmids were also present in isolates. Particularly, plasmid type lncFIB was present in 65 isolates. Additionally, Appendix A shows that the presence or absence of virulence factor genes is different between isolates within the same MLST. The isolates do not cluster per MLST (data not shown), and there is no collinearity observed between MLST and VF genes present in this study sample.

To analyse if our isolate collection is representative, we applied statistical methods estimating its species diversity [14]. Appendix A shows a rarefaction curve with asymptotic behaviour, indicating that most of the virulence factors that are present in this clinical environment were sampled and analysed.

### 3.3. Antibiotic Resistance

Appendix A represents the antibiotic resistance genes that were detected using the ResFinder database. All isolates harboured the *mdf(A)* gene. This is related to macrolide resistance, although macrolides are clinically less important and not used anymore to treat *E. coli* infections. The majority of isolates also harboured the *sitABCD* gene which confers resistance to peroxide as a disinfectant. Resistance genes related to aminoglycosides and sulphonamides were also frequent. Beta-lactam resistance genes were not frequently detected.

Additionally, phenotypic resistance results were available for clinically relevant antibiotics. All blood isolates were routinely tested for amoxicillin, amoxicillin/clavulanic acid, piperacillin and tazobactam, ceftazidime, cefuroxime, cefepime, ceftriaxone, ciprofloxacin and meropenem. Figure 1 shows the number of isolates with phenotypic resistance and with resistance genes detected. In most cases, phenotypic and genotypic resistance had an agreement of 70% or more. Additionally, using the kappa score, there was almost perfect agreement for amoxicillin and meropenem (k = 0.887 and k = 1, respectively) and substantial agreement for piperacillin and tazobactam, ceftazidime, cefepime, and ciprofloxacin (k = 0.771, 0.656, 0.796, and 0.625, respectively). The percentage agreement (73%) and kappa score (k = 0.313, ‘fair’) were lowest for amoxicillin/clavulanic acid, where phenotypically-resistant isolates were classified more often as susceptible in the genotypic analysis. Cephalosporins of the second and third generation were compared per class since the tested antibiotic was different (cefuroxime versus cefoxitin and ceftriaxone versus cefotaxime results). The kappa scores were 0.541 (moderate) for third-generation cephalosporins and 0 (slight) for second-generation cephalosporins.

### 3.4. Virulence Genes

Figure 2 shows the distribution of VF genes identified and the number of VF genes per isolate. Virulence genes *gad* and *terC*, which encode for the glutamate decarboxylase important for acid-resistance and the tellurium ion resistance protein important for tellurite resistance, respectively, were present in all isolates. Both acid and tellurite are frequent additives to growth media. Ninety-two isolates contained *iss* (increased serum survival) and *ompT* (outer membrane protein T), while ninety-one isolates contained *irp2* and *sitA*, all of which encode for important proteins that regulate iron uptake. Isolates had from six to 41 VF genes present.

Figure 3 shows the VF genes per isolate together with clinical outcomes of patients from which these isolates were obtained.

### 3.5. Control Samples

Twenty isolates that were not obtained from blood were selected from the NCBI database. The number of VF genes detected and the VF gene per isolate are shown in Figure 4. Although a similar distribution was observed, several differences were found. In total, 38/98 (39%) VF genes that were detected in blood isolates were not detected in control isolates. Two of these 38 (5%) VF genes were statistically significantly more frequent in blood isolates. These were *mcmA* (Microcin M, which exhibits antibacterial activity) and *sfaD* (S Fimbriae for adhesion to endothelial cells). Out of 77 genes in control isolates, 17 (22%) were not found in blood isolates. Seven of these genes were significantly less frequent. Blood isolates and control isolates had 60 genes in common, of which 11/60 (18%) were significantly different.

### 3.6. VF and Clinical Outcome

#### 3.6.1. Mortality

Only one virulence factor gene, *kpsMII_K23*, a K1 capsule group 2 of *E. coli* type K23, was significantly more present in isolates of patients who died (Table 2). It enables resistance to complement activation and phagocytosis, immunological tolerance, and intracellular survival [1]. Patients with an isolate harbouring *kpsMII_K23* were 7.35 times more likely to die than those without (95%CI: 2.77–19.51, *p* = 0.009). This results in an absolute risk increase of 52%. Based on the sepsis mortality rate of 27%, reported by Rudd et al. [15] and the WHO [13], the absolute mortality risk for patients with isolates with *kpsMII_K23* increases to 41%.

#### 3.6.2. ICU Admission

*kpsMII_K23* (K1 capsule group 2 of *E. coli* type K23) and *cvaC* (Microcin C) genes were significantly more frequent in isolates of patients who were admitted to the ICU (Table 2). This could be in part related to mortality risk, as patients with higher risk are admitted to the ICU. Like all microcins, Microcin C exhibits antibacterial activity and results in increased survival of isolates producing Microcin C. Other studies found that *cvaC* was associated with biofilm-producing UPEC and is frequently more present in isolates from patients with prostatitis or pyelonephritis than in isolates from cystitis [16,17,18]. Indeed, *cvaC* was not found to be significantly different between blood isolates and control isolates here. Patients with an isolate harbouring *kpsMII_K23* were 7.35 times (95%CI: 2.77–19.51, *p* = 0.009) more likely to be admitted to the ICU than those without, while those with isolates harbouring *cvaC* were 3.94 times (95%CI 1.35–11.50, *p* = 0.026) more likely to be admitted to the ICU. This results in an absolute risk increase of 52% and 31%, respectively. Based on the ICU-treated sepsis incidence in the EU of 139 cases per 100,000 persons, reported by the WHO [13], the absolute risk for ICU admission for patients with isolates with *kpsMII_K23* or *cvaC* increases to 0.211% and 0.182%, respectively.

When combining mortality and ICU admission into one outcome parameter (worse outcome) only K1 capsule group 2 was significantly different.

#### 3.6.3. Sepsis

Table 2 presents all 14 VF genes that significantly differed between patients with and without sepsis. *mchB* and *mchC* were significantly more frequent in isolates from patients with sepsis. Both are microcins, of the same gene cluster as *mcmA*, that increase *E. coli* fitness in microbial communities such as the human microbiome [19,20]. The following genes were less frequent in isolates from patients with sepsis: *papA_fsiA_F16, sat, senB, iucC, iutA*, and *iha.* While *papA_fsiA_F16* encodes for the P-fimbriae, the other genes are involved in iron uptake and invasion of the urinary tract or intestines. *sat* has a cytotoxic role in epithelial cells of the bladder and intestines and has a proteolytic effect on complement proteins, resulting in the evasion of the innate immune system. Additionally, *iha* is found on the same pathogenicity island as *sat.* It encodes for an adhesin receptor that is important for kidney and bladder colonization. *senB, iucC* and *iutA* are all part of the iron uptake system, which is necessary for survival in the iron-depleted urinary tract. Additionally, *senB* triggers complement activation and thus the killing of bacteria in the blood [21]. *sfaD*, *cnf1*, *focG*, *vat*, *cldB* and *mcmA* all encode for toxins, except for *sfaD* and *focG*, which encode for the S fimbriae and adhesins, respectively. The S fimbriae, encoded by *sfaD*, binds to endothelial cells, activates plasminogen, generates bacterium-bound plasmin and results in invasiveness/persistence and systemic activation of fibrinolysis. The other genes that are more frequent in sepsis all encode for toxins or enzymes that activate parts of the immune response: *cnf1* activates the NLP3 inflammasome (via Rho GTPase activation), resulting in IL1b production [22]; *vat* is a member of Serine protease autotransporter proteins (SPATE) and induces vacuole-forming in bladder epithelium and loss of intercellular contacts. Last, *cldB* is a key enzyme for production of colibactin, which induces DNA damage and promotes tumour formation and intestinal inflammation.

The relative risk of all virulence factor genes is reported in Table 2. Based on the European sepsis incidence of 289 cases per 100,000, reported by Rudd et al. [15] and the WHO [13], the absolute risk of sepsis increased to 0.351%–0.386% or decreased to 0.223%–0.175%.

#### 3.6.4. Infection Focus

Table 3 presents VF genes that were significantly more different in isolates from different foci of infection. *papC* and *papA_fsiA_F16* both encode for P fimbriae and are found more frequently in isolates from patients with urosepsis. *celb* was significantly more frequent in isolates from patients with bloodstream infections. It encodes a colicin that increases fitness by killing other bacterial cells. *f17A* and *f17G* (fimbriae) were only found in two isolates, but both from patients with bloodstream infections.

### 3.7. BLAST Analysis

BLAST analyses showed that all VF genes in Table 1 and Table 2 have been previously found in other isolates from blood. Moreover, all VF genes were also found in urinary isolates.

## 4. Discussion

This study has identified several VF genes in *E. coli* blood isolates and their association with clinical outcomes of patients presenting with suspected sepsis at a regional emergency department. The K1 capsule group 2 was a major factor associated with worse outcome (mortality and ICU admission). On the other hand, several genes encoding for microcins, toxins and fimbriae were associated with sepsis, while genes involved in iron uptake seemed to be protective of developing sepsis.

Isolates had low levels of antibiotic resistance, and phenotypic resistance was in overall good agreement with genotypic resistance, except for amoxicillin/clavulanic acid. The relatively low level of resistance observed in this study, both geno- and phenotypical, reflects the situation in the community in Belgium, although resistance can be substantial in the healthcare setting [23]. However, the major disagreement seen for amoxicillin/clavulanic acid can be of clinical importance, as this is the first line of treatment. This shows the importance of phenotypic testing in the clinical setting.

Isolates had a wide range of VF genes present, most frequently genes related to iron uptake (*iss, ompT, irp2* and *sitA)*. Moreover, several VF genes that were associated with clinical outcomes of patients were identified. The availability of data regarding clinical outcome of patients, more specifically mortality and ICU admission, in this study has led to the identification of two genes that are associated with worse outcomes. These are the genes *kpsMII_K23* and *cvaC*, which encode K1 capsule group 2 and Microcin C. While Microcin C is an antibiotic that increases *E. coli* fitness in the microbiome, the K1 capsule enables resistance to complement activation and phagocytosis, immunological tolerance, and intracellular survival [24,25,26]. The latter may contribute to the typical dysregulation of the innate immune response seen in sepsis patients, which may result in an uncontrollable disease and death. While several studies reported an association with healthcare-associated infections or with infection focus [24,26], to the best of our knowledge, only one study was identified that found an association with mortality. Mora-Rillo *et al.* found that *fuyA* (yersiniabactin) was associated with increased mortality [27]. This was not found in our study, since *fuyA* was found in 91 out of 103 isolates.

The virulence factors that were found to be related to sepsis in our study are known to be associated with disease severity. Studies have illustrated the relation between VF and disease severity or source of infection. *sat* has previously been found in high frequency in isolates from patients with bacteraemia [28,29,30,31], and *iha* was found in typical sepsis-associated ExPEC ST95 and ST127 B2 strains [32,33,34,35]. Furthermore, it is seen that *senB* was not associated with bacteraemia and was more frequently found in bacteraemia from urinary origin than in bacteraemia from non-urinary origin. *sfaD*, *cnf1*, *focG*, *vat*, *cldB* and *mcmA* were more frequently found in isolates from patients with sepsis. *sfaD* was more frequent in septic and urinary isolates [36,37]. Adhesins from the foc gene cluster (*focG*) were associated with urinary tract colonization and mutations leading to asymptomatic bacteriuria [38]. The cytotoxin encoded by *vat* contributes to UPEC fitness and has been seen in up to 68% of urosepsis isolates. Interestingly, increased *vat* IgG titres were seen in patients [2,39,40,41,42,43]. This seems to be the case in our study as well. Microcins, which create fitness over other bacteria, toxins, which exert damaging effects on the host, and fimbriae, which contribute to bacterial fitness, were more frequently found in patients with sepsis, thus resulting in a worse disease severity. On the other hand, adhesins, which contribute to colonization, and proteins of the iron uptake system seem to have a protective effect. These contribute to bacteraemia but not to sepsis or disease severity.

The correlation of microcins and disease severity could be explained by the protective function of the microbiome. An infection, and its antimicrobial treatment, can result in dysbiosis of the human gut microbiome. The dysbiosis has an additional impact on the host response that, in the case of sepsis, is already aberrated [44,45]. This has also been described for the urinary tract microbiome. The urinary tract microbiome protects against pathogenic colonization. Studies report an increased susceptibility to UTIs with decreased diversity of the urinary tract microbiome [46,47]. Our findings in mostly urogenic *E. coli* can support this. *E. coli* isolates with genes that express antimicrobial peptides and toxins, such as *cvaC, mchB, mchC* and *mcmA*, are more frequent in patients with worse disease. It is likely that pathogenic *E. coli* uses these antimicrobial peptides and toxins to destroy (parts of) the protective microbiome, thereby facilitating colonization and tissue invasion. Additional arguments for this can be found in the comparison with control samples, where a similar distribution was observed, but several differences existed. Mainly, *mcmA* was more frequent in blood isolates than in non-blood isolates, which were mostly of urinary tract origin.

*E. coli* isolated from blood were most frequently ST69, ST73 or ST131 and were of serotype O6:H1, O25:H18 or O1:H7. Indeed, recent studies show that *E. coli* ST131 and ST73 (phylogroup B2) and ST69 were the most frequent in bloodstream infections in the United Kingdom. More specifically, in two studies, these sequence types comprised approximately half of all bloodstream isolates in the UK [48,49]. ST131 was the main type in 20.5% of bloodstream isolates in four centres across Europe (Berlin, Utrecht, Madrid, and Geneva) [50]. These sequence types correspond with the identified serotypes [51].

The strengths of our study are the large number of blood isolates sequenced with WGS and the availability of clinical data of patients. Moreover, previous studies mostly analysed extended spectrum beta-lactamase (ESBL)-producing, multidrug-resistant (MDR) organisms. Since the genetic association between resistance and virulence traits is still poorly understood in these resistant strains [52], the analysis of VF in less resistant *E.coli* isolates from our population allows an assessment of the relative contribution of VF to disease severity.

On the other hand, several limitations exist. First, the absence of a true control group limits the interpretation of contributions of VF genes to outcomes. To try to overcome this, a limited set of publicly available sequences was included. However, the overrepresentation of UTIs and urosepsis, both in our study and in Genbank, has made comparison difficult. Both control isolates and blood isolates seemed to have a similar distribution of VF genes, and all VF genes have also been found before in other non-blood isolates, mostly urinary. Of course, UPEC are the most frequent ExPEC. Additionally, the representation of *iss, ompT, irp2,* and *sitA* in these isolates supports the argument of overrepresentation of UPEC isolates, since iron uptake systems are more frequent in UTIs and urosepsis, because of lower iron availability in the urinary tract [32,33,34,35]. Second, no data are available on a transcriptomic or proteomic level. The presence of these VF genes does not directly translate into the presence of proteins that exert their functions. However, based on the agreement between geno- and phenotypic antibiotic resistance, we could deduce similar levels of agreement for VF. Last, although the data quality is good, with no missing data, this represents a single regional community population. Several biases could have been introduced. For example, the start of the infection is unknown, and patients could have been ill for a longer period before presentation at the ED, which could impact the outcome. Furthermore, the appropriateness of therapy is unknown, and ICU admission is dependent on hospital policy. This impacts the generalizability of these results. Additionally, this study only represents a specific population of community-acquired infection in patients with relatively few comorbidities. Since comorbidities can influence the outcomes, caution should be applied when comparing results with other populations such as ICU patients, immunocompromised patients, or neonates.

In conclusion, *E. coli* isolates from blood cultures of patients with suspected sepsis at the emergency department had a range of different VF genes that contribute to disease severity and clinical outcome. Microcins, toxins, and fimbriae were associated with sepsis and disease severity while adhesins and iron uptake system proteins seemed to be protective. More importantly, two VF genes were found to be associated with worse clinical outcomes. The K capsule group 2 enables resistance to innate immune response and immunological tolerance. Additionally, Microcin C is an antibacterial that increases fitness of *E. coli* over other bacteria and isolates. Both VF genes were significantly more frequent in isolates from patients who died or who were admitted to the ICU. These findings could contribute to a better understanding of host–pathogen interactions. Ultimately, these findings can improve diagnostics to help identify patients most at risk for a worse outcome.

## Figures and Tables

**Figure 1 microorganisms-11-01827-f001:**
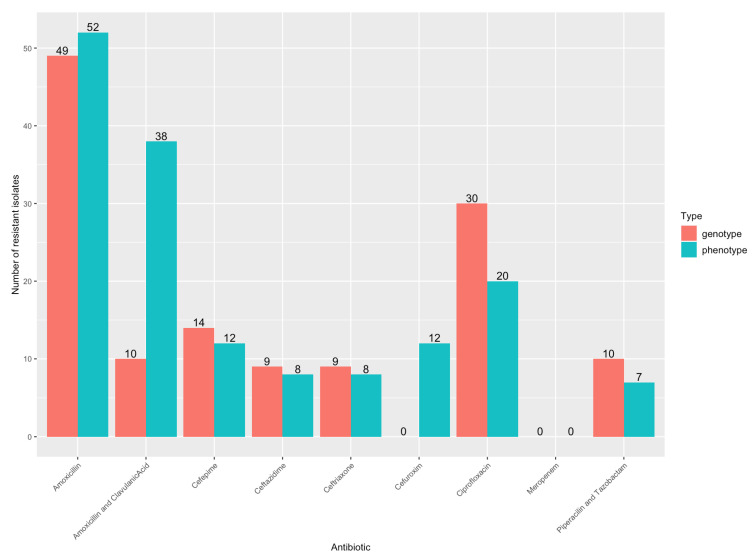
Number of isolates with genotypic and phenotypic antibiotic resistances.

**Figure 2 microorganisms-11-01827-f002:**
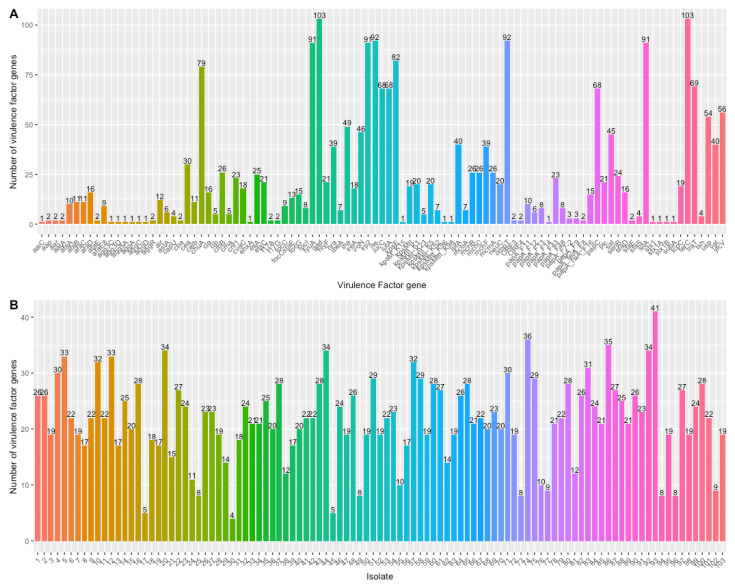
Virulence factor genes identified. (**A**) Virulence factor genes and number detected. (**B**) number of virulence factor genes per isolate.

**Figure 3 microorganisms-11-01827-f003:**
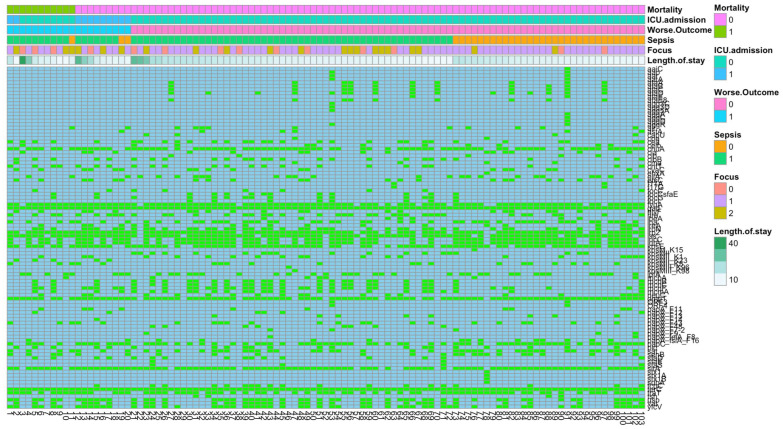
Virulence gene factors and clinical outcomes. Infection focus includes bloodstream, urinary tract, or abdominal tract. Isolates are sorted based on patient outcomes.

**Figure 4 microorganisms-11-01827-f004:**
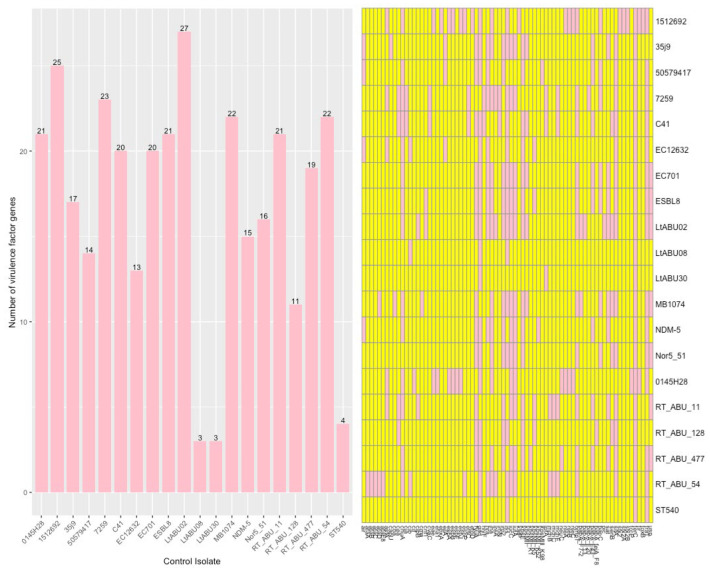
Virulence factor genes in control samples. Left: Number of genes detected. Right: Presence of genes per control isolate in pink.

**Table 1 microorganisms-11-01827-t001:** Diagnoses of infection, SOFA score, ICU admission, mortality.

		Total, *n* = 113 Patients
Age (Median, IQR)		74 (66.5–84)
Sex (female)		54 (52.4%)
**Diagnosis of Infection**		
Secondary BSI		98 (86.7%)
*Urosepsis*		75 (66.4%)
*Intra-abdominal infection*		23 (20.4%)
	Cholangitis	11 (47.8%)
	Cholecystitis	8 (34.7%)
	Peritonitis	1 (4.3%)
	Colitis	1 (4.3%)
	Abscess	1 (4.3%)
	Enterocolitis	1 (4.3%)
Primary BSI		15 (12.4%)
*Endocarditis*		1 (0.8%)
**SOFA score**		
Median (IQR)		2 (1–4)
SOFA = 0		10 (8.8%)
SOFA = 1		22 (19.5%)
SOFA = 2		29 (25.7%)
SOFA = 3		22 (19.5%)
SOFA = 4		16 (14.2%)
SOFA = 5		7 (6.2%)
SOFA = 6		4 (3.5%)
SOFA = 7		1 (0.8%)
SOFA = 8		2 (1.8%)
ICU admission		14 (12.4%)
All-cause in-hospital mortality		11 (9.7%)
Length of stay (Median, IQR)		6 (4–11) days

**Table 2 microorganisms-11-01827-t002:** Number (%) of virulence factor genes from isolates with patients with different outcomes and relative risk (95% CI) of death.

Mortality		Survived	Died	*p*-Value	Relative Risk	*p*-Value
*kpsMII_K23*	Absent	90 (91.8)	8 (8.2)	0.004	7.35 (2.77–19.51)	0.009
	Present	2 (40.0)	3 (60.0)			
**ICU Admission**		**Not Admitted**	**Admitted**	** *p* ** **-Value**	**Relative Risk**	** *p* ** **-Value**
*kpsMII_K23*	Absent	90 (91.8)	8 (8.2)	0.004	7.35 (2.77–19.51)	0.009
	Present	2 (40.0)	3 (60.0)			
*cvaC*	Absent	79 (92.9)	6 (7.1)	0.030	3.94 (1.35–11.50)	0.026
	Present	13 (72.2)	5 (27.8)			
**Sepsis**		**No**	**Yes**	** *p* ** **-Value**	**Relative Risk**	** *p* ** **-Value**
*mchB*	Absent	32 (41.6)	45 (58.4)	0.003	1.58 (1.27–1.97)	0.001
	Present	2 (7.7)	24 (92.3)			
*mchC*	Absent	32 (41.6)	45 (58.4)	0.003	1.58 (1.27–1.97)	0.001
	Present	2 (7.7)	24 (92.3)			
*papA_fsiA_F16*	Absent	24 (27.3)	64 (72.7)	0.007	0.46 (0.22–0.95)	0.005
	Present	10 (66.7)	5 (33.3)			
*sat*	Absent	13 (22.4)	45 (77.6)	0.017	0.69 (0.51–0.93)	0.011
	Present	21 (46.7)	24 (53.3)			
*senB*	Absent	21 (26.6)	58 (73.4)	0.023	0.62 (0.40–0.98)	0.016
	Present	13 (54.2)	11 (45.8)			
*iucC*	Absent	6 (17.1)	29 (82.9)	0.025	0.71 (0.55–0.91)	0.014
	Present	28 (41.2)	40 (58.8)			
*iutA*	Absent	6 (17.1)	29 (82.9)	0.025	0.71 (0.55–0.91)	0.014
	Present	28 (41.2)	40 (58.8)			
*iha*	Absent	12 (22.2)	42 (77.8)	0.025	0.71 (0.53–0.95)	0.016
	Present	22 (44.9)	27 (55.1)			
*sfaD*	Absent	33 (38.0)	54 (62.0)	0.029	1.51 (1.23–1.86)	0.010
	Present	1 (6.3)	15 (94.7)			
*cnf1*	Absent	31 (38.8)	49 (61.2)	0.040	1.42 (1.12–1.80)	0.019
	Present	3 (13.0)	20 (87.0)			
*focG*	Absent	33 (37.5)	55 (62.5)	0.040	1.49 (1.21–1.84)	0.016
	Present	1 (6.7)	14 (93.3)			
*vat*	Absent	26 (41.3)	37 (58.7)	0.043	1.36 (1.05–1.76)	0.026
	Present	8 (20.0)	32 (80.0)			
*clbB*	Absent	30 (39.0)	47 (61.0)	0.048	1.39 (1.09–1.77)	0.026
	Present	4 (15.4)	22 (84.6)			
*mcmA*	Absent	30 (39.0)	47 (61.0)	0.049	1.39 (1.09–1.77)	0.026
	Present	4 (15.4)	22 (84.6)			

**Table 3 microorganisms-11-01827-t003:** Number (%) of virulence factor genes from isolates with patients with different infection foci.

Infection Focus		Bloodstream Infection	Urosepsis	Abdominal Sepsis	*p*-Value
*papC*	Absent	9 (25.7)	15 (42.9)	11 (31.4)	0.001354
	Present	5 (7.4)	53 (77.9)	10 (14.7)	
*papA_fsiA_F16*	Absent	14 (15.9)	53 (60.2)	21 (23.9)	0.01091
	Present	0 (0.0)	15 (100.0)	0 (0.0)	
*celb*	Absent	10 (10.9)	61 (66.3)	21 (22.8)	0.02706
	Present	4 (36.4)	7 (63.6)	0 (0.0)	
*f17A*	Absent	12 (11.9)	68 (67.3)	21 (20.8)	0.001529
	Present	2 (100.0)	0 (0.0)	0 (0.0)	
*f17G*	Absent	12 (11.9)	68 (67.3)	21 (20.8)	0.001529
	Present	2 (100.0)	0 (0.0)	0 (0.0)	

## Data Availability

The data sets used and analysed during the current study are available from the corresponding author on reasonable request. The data are not publicly available due to their containing information that could compromise research participant privacy/consent. Individual participant data that underlie the results reported in this article, after de-identification (text, tables, figures, and appendices), and the study protocol, will be made available upon publication.

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
