# Peer review of "Virulence Factor Genes in Invasive Escherichia coli Are Associated with Clinical Outcomes and Disease Severity in Patients with Sepsis: A Prospective Observational Cohort Study"

_microorganisms, 2023, doi:10.3390/microorganisms11071827_

Round 1
Reviewer 1 Report
The article "Virulence factor genes in invasive Escherichia coli are associ-2 ated with clinical outcomes and disease severity in patients with sepsis: a prospective observational cohort study" is very interesting and suitable for the scopus of the journal. It highlights the number of isolates sequenced and the comparisons in terms of virulence and resistance genes in the different origins of the isolates. Their weakness is the control samples, as detailed in the discussion. However, the rationale for using strains from NCBI databases is convincing. I have a few minor comments which I detail below.
Line 75: please eliminate the extra space between words.
Line 159: the concept of flora to refer to the microbiota is currently incorrect. Please correct it.
Line 381: please correct the position of the comma.
Line 392: correct the word "wes".
Line 393: was or was no?..I understand that the correct is "was". Revise it.
Lines 395-396: please revise the sentence. It is not clear.
Line 402: eliminate the extra "punctuation".
Lines 424-437: please revise the paragraph.I see E. coli not correctly written.
Author Response
We thank the reviewer for their time and comments. All comments have been revised and adapted in the manuscript. No changes were made for comment 2, since the sentence describes the definition used during the study for assigning patients with true bacteremia and those with contaminated blood cultures. This does not refer to the microbiome.
Reviewer 2 Report
Thank you for asking me to review this interesting paper which deals with a very important topic such as the impact of bacterial virulence factors on septic patients' mortality. There are very few papers published on this topic and the submission is. therefore, relevant.
There are few similar studies conducted in the neonatal population showing different results. I suggest the authors to add a short section in the discussion comparing their data with other similar studies conducted on different pathogens or in different populations.
Author Response
We thank the reviewer for their time to revise and suggestions. Indeed, there are few but some relevant studies in other populations. However, it was chosen not to include these because comparisons are often difficult between these populations and ours. The population in this study are patients admitted to the emergency department. They thus reflect the community and are mostly not critically ill at inclusion, as compared to an ICU population. Furthermore, our study population has relatively few comorbidities. Of course, comorbidities play a more important role, thus probably also influencing other studies’ results. We have included this explanation in brief in the discussion Line 471. We feel that by providing a thorough comparison and literature search as is now provided, would be sufficient to interpret these results.
Reviewer 3 Report
While I enjoyed your account of comparing virulence factors found against outcomes I do have some reservations concerning some of your findings, especially the emphasis on KpsMII_K23 as being of particular importance; what other VFs were in these isolates? Are there sufficient numbers to infer statistical significance? Was there something about those patients or their treatment that perhaps played a part in their poor outcome? Other concerns are:
lines 24,150
what is a SOFA score?
Table 1
Your percentages add up to more than 100 - presumably because some patients appear in more than one group?
line 222
ST73 is strongly associated with virulence factors, which in my experience are found consistently in that type. In lines 225-8 you say 'Additionally, supplementary figure 2 shows that the presence or absence of virulence fac-tor genes is different between isolates within the same MLST. The isolates do not cluster per MLST (data not shown), and there is no collinearity observed between MLST and VF genes present.' I find that extremely surprising for ST 73 which in my experience is consistently associated with multiple virulence factors (as are also STs 12, 80 and 998).
I find Supplementary Figure 2 difficult because many STs share highly similar (?identical) colours so it is difficult to relate VFs with a particular ST (e.g. many STs are represented by pink squares and I can't work out which column relates to each one.)
line 224
Did these lncFIB plasmids carry resistance genes?
Surely outcome will also depend on whether effective and appropriate antibiotic therapy was given, but this is not mentioned. Do you have any comments or data on that?
I worry about your emphasis on kpsMII_K23. There may be other reasons why those patients in particular died. Did these isolates carry multiple VFs? Were they particularly resistant? We need to know a bit more about them to enable assessment of the importance of this VF.
Author Response
We thank the reviewer for their time and thorough review. Please find our changes and answers below.
